# Targeting Oxidative Phosphorylation Reverses Drug Resistance in Cancer Cells by Blocking Autophagy Recycling

**DOI:** 10.3390/cells9092013

**Published:** 2020-09-01

**Authors:** Jae-Seon Lee, Ho Lee, Hyonchol Jang, Sang Myung Woo, Jong Bae Park, Seon-Hyeong Lee, Joon Hee Kang, Hee Yeon Kim, Jaewhan Song, Soo-Youl Kim

**Affiliations:** 1Division of Cancer Biology, Research Institute, National Cancer Center, Goyang, Gyeonggi-do 10408, Korea; ljs891109@gmail.com (J.-S.L.); hjang@ncc.re.kr (H.J.); shlee1987@gmail.com (S.-H.L.); wnsl2820@gmail.com (J.H.K.); 74790@ncc.re.kr (H.Y.K.); 2Department of Biochemistry, College of Life Science and Biotechnology, Yonsei University, Seoul 03722, Korea; 3Department of System Cancer Science, Graduate School of Cancer Science and Policy, National Cancer Center, Goyang, Gyeonggi-do 10408, Korea; ho25lee@ncc.re.kr (H.L.); jbp@ncc.re.kr (J.B.P.); 4Center for Liver and Pancreatobiliary Cancer, National Cancer Center, Goyang, Gyeonggi-do 10408, Korea; wsm@ncc.re.kr

**Keywords:** cancer metabolism, energy metabolism, aldehyde dehydrogenase, oxidative phosphorylation (OxPhos), ATP production

## Abstract

The greatest challenge in cancer therapy is posed by drug-resistant recurrence following treatment. Anticancer chemotherapy is largely focused on targeting the rapid proliferation and biosynthesis of cancer cells. This strategy has the potential to trigger autophagy, enabling cancer cell survival through the recycling of molecules and energy essential for biosynthesis, leading to drug resistance. Autophagy recycling contributes amino acids and ATP to restore mTOR complex 1 (mTORC1) activity, which leads to cell survival. However, autophagy with mTORC1 activation can be stalled by reducing the ATP level. We have previously shown that cytosolic NADH production supported by aldehyde dehydrogenase (ALDH) is critical for supplying ATP through oxidative phosphorylation (OxPhos) in cancer cell mitochondria. Inhibitors of the mitochondrial complex I of the OxPhos electron transfer chain and ALDH significantly reduce the ATP level selectively in cancer cells, terminating autophagy triggered by anticancer drug treatment. With the aim of overcoming drug resistance, we investigated combining the inhibition of mitochondrial complex I, using phenformin, and ALDH, using gossypol, with anticancer drug treatment. Here, we show that OxPhos targeting combined with anticancer drugs acts synergistically to enhance the anticancer effect in mouse xenograft models of various cancers, which suggests a potential therapeutic approach for drug-resistant cancer.

## 1. Introduction

Resistance to chemotherapeutic drugs is a leading cause of death from cancer and, therefore, a major focus in cancer research. Many mechanisms of drug resistance have been proposed based on the expression of a unique set of genes that dictate tumor progression under chemotherapy [1,2]. Targeting the signaling pathways of the genes and enzymes associated with biosynthesis is unlikely to lead to a method to overcome drug resistance in cancer because such therapeutics can elicit various rerouting pathways in tumor cells [3,4]. 

Recently, autophagy enhancement in response to chemotherapy has been considered as a major mechanism of drug resistance [5]. In drug resistance cases, autophagy inhibition in combination with chemotherapy can significantly increase cancer cell death, which supports the theory that autophagy induces tumor survival by contributing to drug resistance against chemotherapy [5]. However, autophagy can occur as either a prodeath or prosurvival response to multiple stresses in normal and cancer cells [6,7]. Hence, autophagy is suggested as a therapeutic target for overcoming anticancer drug resistance, although how the level of autophagy could be changed to achieve this remains under investigation [8,9].

Cytotoxic cancer drugs, such as irinotecan, inhibit the DNA biosynthesis of cancer cells by increasing DNA instability through the inhibition of both DNA replication and transcription [10]. Autophagy is triggered by the inhibition of biosynthesis, rerouting a supply of recycled amino acids for biosynthesis required for cell survival [11]. Irinotecan-resistant non-small cell lung cancer (NSCLC) cells show an increase in autophagy [11,12], as well as in mitochondrial oxidative phosphorylation (OxPhos), compared with drug-unexposed (wild-type) NSCLC cells [12]. Irinotecan treatment increases the level of electron transfer complexes of the OxPhos pathway in the mitochondrial membrane, which allows cancer cells to produce more adenosine triphosphate (ATP) [12]. This suggests that cancer cells depend on ATP production from OxPhos to overcome cytotoxic damage. 

These findings suggest that targeting tumor OxPhos could be a way to reverse drug resistance. This is a realistic possibility because the OxPhos complex, as a catabolic process in cancer, can be selectively differentiated from that in normal cells, as it is supplied with NADH from a different source [13]. Electron supply for ATP production in OxPhos of cancer cells largely depends on cytosolic NADH produced by dehydrogenases, such as aldehyde dehydrogenase (ALDH), while NADH production in normal cells depends on the tricarboxylic acid cycle (TCA cycle) [13,14,15]. Therefore, targeting OxPhos in cancer cells by inhibiting ALDH to reduce NADH production could selectively reduce the ATP level, causing selective inhibition of autophagy, leading to selective cancer cell death. By contrast, in normal cells, autophagy may not be affected by the inhibition of ALDH because ATP is produced from OxPhos using NADH mainly supplied from the TCA cycle [13]. 

Here, we tested whether the inhibition of OxPhos using inhibitors of ALDH and of the mitochondrial membrane electron transfer complex component I has a sensitizing effect on anticancer drug resistance by triggering autophagic cell death in vitro and in a mouse xenograft model, using cell lines derived from a variety of cancers.

## 2. Materials and Methods

### 2.1. Cell Culture

Human cancer cell lines were obtained from the American Type Culture Collection (ATCC) and the Korean Cell Line Bank. All cells were incubated at 37 °C and maintained in 5% CO_2_. A375 (melanoma) and MIA PaCa-2 (pancreatic cancer) were grown in high glucose DMEM (SH30243.01; Hyclone, Logan, UT, USA) containing fetal bovine serum (FBS; SH30070.03HI, HyClone), penicillin, and streptomycin. HT-29 and Colo 205 (colon cancer), UACC-62 (melanoma), SK-OV-3 and OVCAR 3 (ovarian cancer), PC-3 and DU 145 (prostate cancer), SNU-638 (stomach cancer), AsPC-1 (pancreatic cancer), and SNU-449 and Huh-7 (liver cancer) cell lines were grown in RPMI 1640 medium (SH30027.01, HyClone) containing FBS, penicillin, and streptomycin. The AGS (stomach cancer) cell line was grown in ATCC-formulated F-12K medium (30-2004; ATCC, Manassas, VA, USA) containing FBS, penicillin, and streptomycin.

### 2.2. Sulforhodamine B-Cell Proliferation Assay

Cell suspensions (100 μL) were plated in 96-well microtiter plates at densities ranging from 5–20 × 10^3^ cells/well, depending on the doubling time of the cell line. The plates were then incubated for 24 h prior to the addition of the experimental drugs. The drugs were prepared at the appropriate concentrations, i.e., 100 μL was added to each well, and the plates were incubated at 37 °C in a 5% CO_2_ incubator. The assay was terminated by the addition of cold trichloracetic acid (TCA). The cells were fixed in situ by gently adding 50 μL cold 50% (*w/v*) TCA (final concentration, 10% TCA), and were incubated for 60 min at 4 °C. The supernatant was discarded, and the plates were washed five times with tap water and then air dried. Sulforhodamine B (SRB) solution (100 μL) at a concentration of 0.4% (*w/v*) in 1% acetic acid was added to each well, and the plates were then left for 10 min at room temperature. After staining, the unbound dye was removed by washing five times with 1% acetic acid; the plates were then air dried. The bound stain was subsequently solubilized with 10 mM Tris base, and the absorbance was recorded using an automated plate reader at 515 nm.

### 2.3. Western Blot Analysis

The cells were harvested, washed in phosphate-buffered saline (PBS), and lysed in lysis buffer (20 mM Tris-HCl (pH 7.4), 150 mM NaCl, 1% (*v*/*v*) Triton X-100, 1 mM EDTA, protease, phosphatase inhibitor cocktail). The protein concentration of the cell lysates was quantified using a BCA Protein Assay Kit (Pierce, Rockford, IL, USA). The protein samples were subjected to 6–15% sodium dodecyl sulfate-polyacrylamide gel electrophoresis and transferred onto polyvinylidene difluoride membranes. After blocking with 5% bovine serum albumin (BSA), the membranes were incubated with the primary antibodies (see below) diluted in 5% BSA buffer overnight or for 1 h at 4 °C and then with horseradish peroxidase-conjugated secondary antibody for 1 h at room temperature. Finally, the protein band images were captured using a FUSION Solo system (VILBER, Collégien, France) with enhanced chemiluminescent (ECL) reagent (LF-QC0101; Ab FRONTIER, Seoul, Korea). Protein band images were quantified using ImageJ software (64-bit Java 1.8.0_112). The primary antibodies used in the experiments were anti-Total OXPHOS Human WB Antibody Cocktail (ab110411) and anti-Phospho-eIF4B (Ser422) (ab134138; both purchased from Abcam, Cambridge, UK). Anti-eIF4B (sc-3909) and anti-β-actin (sc-47778) were purchased from Santa Cruz Biotechnology (Dallas, TX, USA). Anti-mTOR (#2983S), anti-Phospho-mTOR (Ser2448) (#2971S), anti-4E-BP1 (#9644S), anti-Phospho-4E-BP1 (Thr37/46) (#2855S), anti-Phospho-4E-BP1 (Thr70) (#9455S), anti-p70 S6K (#2708S), anti-Phospho-p70 S6K (Ser371) (#9208S), and anti-Phospho-p70 S6K (Thr389) (#9205S) were purchased from Cell Signaling Technology (Danvers, MA, USA).

### 2.4. Mitochondrial Function Analysis

A Seahorse XF Cell Mito Stress Test Kit (Agilent Technologies, Santa Clara, CA USA) was used to determine cellular oxygen consumption rate (OCR). Cells were incubated in XF base medium supplemented with 10 mM glucose, 1 mM sodium pyruvate, and 2 mM l-glutamine, and were equilibrated in an ambient air incubator for 1 h before starting the assay. The samples were mixed for 3 min and measured for 3 min using an XFe96 Extracellular Flux Analyzer (Seahorse Bioscience). Oligomycin (0.75 μM), carbonyl cyanide-4 (trifluoromethoxy) phenylhydrazone (FCCP) (1 μM), and rotenone/antimycin A (0.5 μM) were injected at the indicated time points. Finally, the OCR data were normalized to account for the differences in the initial cell numbers for each well by using the SRB assay.

### 2.5. Measurement of Mitochondrial Membrane Potential (∆ψm)

Cells were cultured for 24 h in 100 mm dishes or LAB-TEK II 4-well chambered coverglass (Thremo Fisher Scientific, Waltham, MA, USA) and then treated with drugs as indicated. Twenty minutes prior to the end of each treatment, 100 nM tetramethylrhodamine-ethylester (TMRE; ab113852, Abcam) was added to the culture medium. Cells were washed three times with ice-cold PBS. TMRE labeling of the mitochondria membrane potential was analyzed using a FACSCalibur flow cytometer (BD Falcon, Bedford, MA, USA) and an LSM780 Laser Scanning Microscope (Carl Zeiss, Oberkochen, Germany).

### 2.6. Determination of Autophagy 

The autophagy assay was performed by flow cytometry using a CYTO-ID Autophagy Detection Kit (Enzo Life Sciences, Farmingdale, NY, USA) according to the manufacturer’s protocol. Cells were treated with the drug for the indicated duration. At the end of the treatment, samples should have contained 1 × 10^5^ to 1 × 10^6^ cells/mL. Cells were pelleted by centrifugation at 1000 rpm for 3 min, washed in 1× Assay Buffer, and then resuspended in 250 μL of 1× diluted CYTO-ID green stain solution (1 mL 1× Assay Buffer + 2 μL CYTO-ID Autophagy Detection Reagent). After incubation in the dark for 30 min at room temperature or 37 °C, the autophagy levels of the cells were analyzed using a FACS Calibur flow cytometer (BD Falcon). 

Cells were also cultured for 24 h in LAB-TEK II 4-well chambered coverglass (Thermo Fisher Scientific, Waltham, MA, USA) and treated with drugs as indicated. And Medium was removed. Cells were washed in 1× assay buffer, and then stained in 500 μL of 1× diluted CYTO-ID green stain solution and Hoechst33342 (Thremo Fisher Scientific). The cells were incubated for 30 min at 37 °C and maintained in 5% CO_2_. The samples were analyzed using an LSM780 confocal microscope.

### 2.7. Immunohistochemistry

Formaldehyde (4%)-fixed specimens were paraffin-embedded and cut at a thickness of 4 μm. Sections were dried for 1 h at 56 °C, and immunohistochemical staining was performed using a DISCOVERY XT automated staining platform (Ventana Medical Systems, Tucson, AZ, USA) with a DISCOVERY ChromoMap DAB (3,3’-diamniobenzidine) Kit as follows: Sections were deparaffinized and rehydrated using EZ Prep (Ventana) and washed with Reaction Buffer (Ventana), and antigens were retrieved by heating at 90 °C for 30 min in RiboCC citrate buffer reagent (pH 6.0; Ventana) prior to detection with an anti-Ki-67 antibody (ab15580, Abcam) or anti-NDUFB8 antibody (ab192878, Abcam) and DAB.

### 2.8. Xenograft Tumor Models

BALB/c-nude mice (Orientbio, Seongnam, Korea), aged 6–8 weeks, were used for the xenograft model. This study was reviewed and approved by the Institutional Animal Care and Use Committee of the National Cancer Center Research Institute, which is an Association for Assessment and Accreditation of Laboratory Animal Care International (AAALAC International) -accredited facility that abides by the Institute of Laboratory Animal Resources guide (protocols: NCC-18-435, NCC-20-558). The details, including the tumor cell line, drugs, dose, timings, and delivery used for each xenograft mouse model, are given in S1. Cancer cells (5 × 10^6^~1 × 10^7^) in 100 μL PBS were inoculated subcutaneously. The mice were divided into four treatment groups: 1. Control group treated with vehicle (5% dimethyl sulfoxide (DMSO), 5% Cremophor in PBS) only; 2. Gossypol + Phenformin group; 3. Anticancer drugs group; and 4. Anticancer drugs + Gossypol + Phenformin group. Mice received 200 μL treatment. Irinotecan (20 or 40 mg/kg body weight) was administered by intraperitoneal injection 1 or 2 days/week. Vemurafenib (30 mg/kg body weight) was administered orally 5 days/week. Cisplatin (4 mg/kg body weight) was administered by intraperitoneal injection 2 days/week. Doxorubicin (5 mg/kg body weight) was administered orally 5 days/week. Primary tumor size was measured weekly using calipers. Tumor volume was calculated using the formula V = (A × B^2^)/2, where V is the volume (mm^3^), A is the long diameter, and B is the short diameter. Finally, tumor weight was measured by scale at the end point.

### 2.9. Establishment of Resistant Cell

The resistant SNU-638 and MIA PaCa-2 were generated by exposure to the drugs for twenty cycles. SNU-638 and MIA PaCa-2 were respectively treated with 25 nM and 100 nM of Irinotecan for 72 h. For twenty cycles, the cells were selected by serial increase of irinotecan treatment in the culture media. This development period was carried out for approximately 2 months [16].

### 2.10. Statistical Analysis

Statistical analysis was performed using the Student’s *t*-test. Tumor growth in the xenograft mouse model was analyzed statistically by two-way analysis of variance (ANOVA) using Microsoft Excel. A *p*-value ≤0.05 was considered statistically significant.

## 3. Results

### 3.1. Autophagy and Mitochondrial OxPhos Markedly Increases in Drug-Resistant Cancer Cells

Irinotecan-resistant lines of SNU-638 and MIA PaCa-2 were established by long-term treatment with irinotecan; resistance was confirmed using the SRB cell proliferation assay (Appendix A). To test whether autophagy level is related with drug resistance, the autophagy level was measured using the Cyto-ID autophagy specific staining method. Compared with the wild-type counterpart, the irinotecan-resistant lines of SNU-638 and MIA PaCa-2 showed an increase in autophagy levels of 92% and 81%, respectively (Figure 1A), an increase in basal OCR of about 38% and 54%, respectively, (Figure 1B), and an increase in ATP level of about 46% and 29%, respectively (Figure 1B). The drug-resistant lines of SNU-638 and MIA PaCa-2 also showed a 5.8- and 3.1-fold increase in the level of mitochondrial complex I, respectively, compared with the wild-type cell lines (Figure 1C). To test whether the mitochondrial membrane potential of the drug-resistant cells was higher than that of the wild-type cell lines, TMRE was employed as a mitochondrial membrane potential-dependent fluorescent indicator dye. The mitochondrial membrane potential in the irinotecan-resistant lines of SNU-638 and MIA PaCa-2 was increased by 41% and 40%, respectively, compared with the wild-type cell lines (Figure 1D). 

To test whether elevated autophagy and OxPhos had been acquired, levels of autophagy and OCR as an OxPhos activity were measured with a Cyto-ID autophagy detection kit and by XFe96 extracellular flux analysis in the wild-type cell lines, and then after anticancer drug treatment for 24–48 h (Figure 2 and Appendix A). The cells surviving after anticancer drug treatment showed levels of autophagy that increased over time by 1.7-fold and 5.8-fold after 48 h in SNU-638 and MIA PaCa-2, respectively (Figure 2A and Appendix A). Anticancer drug-treated SNU-638 cells also had an increased OCR and ATP level, i.e., up to 2.4-fold and 2.6-fold, respectively, at 48 h compared with untreated cells (Figure 2B and Appendix A). The expression level of mitochondrial OxPhos complexes and the mitochondrial membrane potential were analyzed in cancer cells treated with or without irinotecan (Figure 2C,D and Appendix A). The level of mitochondrial complex I was increased 2.9-fold and 4.9-fold by 48 h in treated SNU-638 and MIA PaCa-2, respectively, while complex II was not increased (Figure 2C). This suggests that cancer cells promote electron entry gate through mitochondrial complex I using NADH, instead of via mitochondrial complex II using FADH_2_, when treated with the anticancer drug. The mitochondrial membrane potential was also increased in the treated SNU-638 and MIA PaCa-2 by 24% and 83%, respectively (Figure 2D and Appendix A). Thus, drug-treated cancer cells showed increased levels of autophagy and OxPhos compared with the wild-type cancer cells. Furthermore, the results indicate that autophagy and mitochondrial OxPhos activity can be induced by anticancer drug treatment.

### 3.2. OxPhos Inhibition by Gossypol and Phenformin Reverses Anticancer Drug Resistance

OxPhos inhibition using inhibitors against mitochondrial complex I and ALDH is known to promote ATP depletion in cancer cells [12,17,18]. Treatment with either the mitochondrial complex I inhibitor phenformin or the ALDH inhibitor gossypol caused only modest tumor regression in a mouse xenograft model, but in combination, they synergized, causing both marked tumor regression and a decrease in ATP production [18]. Instead of gossypol treatment, the combination of the loss of ALDH1L1 deletion and phenformin treatment decreased tumor growth in an in vivo KRAS-driven lung cancer model, and the synergy correlated with a decrease in ATP production [19]. We, therefore, tested whether targeting OxPhos with gossypol and phenformin could reduce the levels of cellular ATP, thereby decreasing irinotecan-resistant SNU-638 and MIA PaCa-2 cell proliferation. The combination of phenformin with gossypol treatment caused a synergistic decrease in cell proliferation (Figure 3A). Next, we tested the effect of blocking OxPhos using phenformin and gossypol on the OCR of wild-type and irinotecan-resistant SNU-638 and MIA PaCa-2. Mitochondrial respiration analysis using a Seahorse XF Cell Mito Stress Test revealed that the OCRs of irinotecan-resistant SNU-638 and MIA PaCa-2 were increased 1.86-fold and 3.13-fold above the basal OCR, respectively, compared with the wild-type counterparts (Figure 3B). The combination of phenformin and gossypol decreased the elevated OCR of irinotecan-resistant SNU-638 and MIA PaCa-2 by 91% and 77%, respectively (Figure 3B). Having shown using TMRE staining that, compared with the wild-type cells, the mitochondrial membrane potential was increased in irinotecan-resistant SNU-638 and MIA PaCa-2 cells (Figure 1D), we found that the treatment of irinotecan-resistant cells with the combination of phenformin and gossypol decreased the elevated mitochondrial membrane potential by 67% and 26%, respectively (Figure 3C). These results indicate that OxPhos inhibition with a combination of phenformin and gossypol decreased OxPhos activity significantly in irinotecan-resistant cells. 

### 3.3. OxPhos Inhibition Promotes Chemotherapy Efficacy in Various Cancer Cells

We observed that anticancer drug treatment killed cancer cells, but that some cells survived through autophagy induction by OxPhos activation, and that the surviving cancer cells later acquired drug resistance. Therefore, the anticancer effect of primary therapeutic drugs may be potentiated with OxPhos inhibitors such as phenformin and gossypol. To test this hypothesis, we used cell lines derived from cancers of the colon (HT29, Colo205), skin (UACC62, A375), ovary (SK-OV-3, OVCAR3), prostate (PC3, DU145), stomach (SNU-638, AGS), pancreas (MIA PaCa-2, AsPC-1), and liver (SNU449, Huh7; Figure 4A and Appendix A). Each cancer type was treated with clinically appropriate primary anticancer drugs, such as irinotecan, vemurafenib, cisplatin, and doxorubicin (Figure 4A and Appendix A). The combination of phenformin and gossypol with the primary anticancer drugs showed a significant synergistic anticancer effect in the 14 cell lines representative of seven different types of cancers (Figure 4A and Appendix A). A morphology indicative of cell death was observed after combination treatment with phenformin and gossypol with irinotecan. SNU-638 and MIA PaCa-2 cells treated with the anticancer drug together with the OxPhos inhibitors showed an increase in intracellular granules and swelling (S3H). These features of cell death were more similar to those of autophagic death than of nonlytic apoptosis. This suggests that OxPhos inhibition using phenformin and gossypol blocks autophagy-mediated survival in cancer cells, which appears to result in death by autophagy. To test whether the phenformin and gossypol combination treatment regulates the mitochondrial membrane potential of anticancer drug-resistant cells without affecting autophagy, we analyzed the mitochondrial membrane potential and autophagy by TMRE and Cyto-ID staining, respectively, (Figure 4B). Compared with the wild-type cells, an increase in autophagy and the mitochondrial membrane potential was observed in irinotecan-resistant SNU-638 cells (Figure 4B). OxPhos inhibition by treatment with the phenformin and gossypol combination for 6 h caused a marked reduction in the mitochondrial membrane potential without a decrease in autophagy in the irinotecan-resistant SNU-638 cells (Figure 4B). This suggests that under drug treatment, drug resistance is acquired in wild-type cancer cells (such as SNU-638) by autophagy induction with an increase in OxPhos. To test whether OxPhos inhibition also regulates the mitochondrial membrane potential without affecting autophagy activity induced by drug treatment, we analyzed the mitochondrial membrane potential and autophagy by TMRE staining and by Cyto-ID staining, respectively, after treatment with irinotecan for 48 h and OxPhos inhibition for 6 h (Figure 4C). Irinotecan treatment significantly increased the mitochondrial membrane potential and autophagy in SNU-638 cells (Figure 4C). OxPhos inhibition with the combination of phenformin and gossypol reduced the mitochondrial membrane potential without affecting the autophagy level (Figure 4C). This implies that autophagy recycling must be completed by mTOR activation with nutrients and ATP supplied from autophagy. It is ironic that autophagy activates mTORC1, because mTORC1 inactivation induces autophagy. The results shown in Figure 1 and Figure 2 support the notion that mTORC1 must be activated in drug-resistant cancer cells and in cancer cells under drug treatment. Using immunoblotting, we tested whether mTORC1 was indeed activated after anticancer drug treatment for 48 h when autophagy was induced (as seen in Figure 4C). Surviving SNU-638 cells under irinotecan treatment showed induction of mTOR activation with levels of p-mTOR, p-p70S6K, p-4E-BP1, and p-eIF4B increasing 1.53-, 1.92-, 4.39-, and 12.5-fold, respectively (Figure 4D). The increased activity of mTORC1 was reversed by OxPhos inhibition using phenformin and gossypol to the level found in the untreated condition (Figure 4D). These results suggest that anticancer drug treatment induces autophagy to promote survival through the recycling of amino acids and energy release. However, it appears that the autophagy-mediated survival pathway is dependent on increased OxPhos for mTORC1 activation, because mTORC1 activation induces autophagy recycling. Therefore, ATP depletion by OxPhos inhibition may reverse drug resistance. 

### 3.4. Anticancer Drug Treatment Combined with Phenformin and Gossypol Suppress the Growth of Tumors in a Mouse Xenograft Model

Having found that OxPhos inhibition had a synergistic antiproliferation effect with anticancer drugs in various cancer cells (Figure 4A), we next investigated the potential of this effect as an anticancer treatment in vivo. To investigate the synergy of OxPhos inhibition using phenformin and gossypol with anticancer drugs, we employed a mouse xenograft model with cancer cell lines derived from colon (Colo 205), skin (A375), ovary (SK-OV-3), prostate (PC-3), stomach (SNU-638), and pancreatic (MIA PaCa-2) cancers (Figure 5). Colo 205 (Figure 5A,B), A375 (Figure 5C,D), SK-OV-3 (Figure 5E,F), PC-3 (Figure 5G,H), SNU-638 (Figure 5I,J), and MIA PaCa-2 (Figure 5K,L) cell lines were injected subcutaneously (near the scapulae) into 6–8-week-old female nude BALB/c mice. We measured the volume of the xenograft tumors weekly using calipers following treatment with anticancer drugs with or without OxPhos inhibitors. OxPhos inhibitors acted synergistically with the anticancer drug treatment to decrease tumor size (Figure 5A,C,E,G,I,K) and final weight (Figure 5B,D,F,H,J,L). As a measure of proliferative activity in the xenograft tumors, we examined the expression of Ki-67 at the end point. The tumor Ki-67 index of xenografted SNU-638 and A375 was decreased by 72% and 70%, respectively, in the OxPhos inhibition group with drug treatment, while in the SNU-638 and A375 drug-only treatment group, the Ki-67 index was decreased by about 20% and 15%, respectively, compared with that in untreated SNU-638 and A375 controls (Figure 6A). In line with the marked increase in mitochondrial complex I under drug treatment observed in vitro (Figure 1 and Figure 2), tumor tissue from the anticancer drug-treated group showed approximately 5- (SNU-638) and 13-fold (A375) increases in expression of NADH dehydrogenase 1 beta subcomplex subunit 8 (NDUFB8) as a mitochondrial complex I compared with the untreated controls (Figure 6B).

## 4. Discussion

Decreasing cellular ATP content induces accumulation in either the G1 or G2-M phase of the cell cycle, depending on the degree of ATP reduction, which is followed by the induction of cell death [20]. In normal cells, depleting ATP by 15% causes death by necrosis, but depletion by 25–70% causes death by apoptosis [21]. In cancer cells, depleting ATP by about 80% by OxPhos inhibition with phenformin and gossypol also induces a delay in cell cycle progression at the G1/S transition within 24 h [17], followed by cell death after 48 h [12,17,18,22]. The cell death induced by ATP depletion is not related to the Bcl-2 level [17], but rather, to autophagy [12]. However, the mechanism by which ATP depletion causes cell death when autophagy is induced by anticancer drug treatment is not known. Here, we found that drug-resistant cancer cells (those surviving anticancer drug treatment) show induction of the autophagy process with mTOR activation accompanying an increase in ATP production as a result of increased OxPhos. This mTORC1 activation through nutrient supply from autophagy allows cancer cell growth to proceed under anticancer drug conditions that are blocking de novo synthesis. Therefore, cutting off ATP production by inhibiting OxPhos can reverse drug resistance in cancer cells.

Cancer cells have higher OxPhos activity than normal cells, accompanied by more cytosolic NADH [15], largely produced by ALDH [17,18], and more mitochondrial complex I as an entry of NADH [12]. Therefore, in the present study, we used phenformin to block OxPhos by inhibition of the mitochondrial electron transport chain (ETC), as well as gossypol to block ALDH activity [23,24]. Phenformin targets mitochondrial complex I in the mitochondrial ETC [25]. ETC reactions generate a proton gradient across the mitochondrial inner membrane by transfer of electrons across molecules such as complex I, III, and IV producing ATP via ATP synthase. The pan-ALDH inhibitor gossypol is a polyphenolic compound that can be made synthetically or produced inexpensively on a very large scale by extraction from cottonseed. Gossypol is a non-competitive inhibitor against substrate in ALDH reactions, but a strong competitive inhibitor against cofactors such as NAD^+^ [23,24]. Previously, we have shown an anticancer effect of gossypol and/or phenformin in xenograft models of glioblastoma [26], NSCLC [18,19], and gastric cancer [22].

In the present study, our results were concordant with previously published data showing that anticancer drug treatment induces an increase in autophagy in cancer cells [5]. We found that drug- resistant cancer cells showed a higher level of autophagy than wild-type cancer cells. This suggests that anticancer drug treatment in cancer cells induces autophagy for survival through the recycling of amino acids. To complete autophagy recycling, cancer cells require mTOR activation for biosynthesis. In this study, we found that wild-type cancer cells die via apoptosis and necrosis, while drug-resistant cells survive through autophagy accompanied by mTOR activation. mTOR can be activated by growth factors and nutrients such as amino acids and ATP [27]. The major role of autophagy is to provide nutrients by degrading cellular contents [28]. It has been reported that the degradation of macromolecules and release of intracellular substituents following autophagy trigger the reactivation of mTOR, which stimulates the recycling of proto-lysosomal membrane components via tubules and then vesicles that mature into a new lysosome [29]. Autophagy-mediated restoration of mTORC1 induces autophagy termination and reformation to lysosomes, which completes the feedback loop [29,30]. We found that cancer cells avoid autophagic cell death by mTOR activation through nutrient replenishment, such as with amino acids [29], which can be stalled by ATP depletion via blocking of OxPhos.

We have shown that treatment with various anticancer drugs increases the OCR and levels of OxPhos electron transfer complexes, which correlates with an increase in NADH and ATP production in cancer cells. These results are also consistent with those of previous reports examining irinotecan treatment in an NSCLC model [12]. ATP depletion by OxPhos inhibition using phenformin and gossypol, therefore, inactivates mTOR, which leads to drug-resistant cancer cell death by blocking autophagy.

## 5. Conclusions

We observed that anticancer drug treatment triggers autophagy, which requires more ATP through the activation of OxPhos for cancer cell survival. We found that anticancer drugs induce the autophagy process for supplying nutrients, such as amino acids and ATP, which then activates the mTOR pathway, promoting cancer growth through biosynthesis. The autophagy process with mTOR activation leads to extended cancer cell survival, which is stabilized, leading to drug resistance. However, drug resistance can be reversed by blocking OxPhos because mTOR activity can be stalled by ATP depletion.

Here, we demonstrated that anticancer drug-induced autophagy leads to autophagic cell death by ATP depletion, and thus, that anticancer drug resistance can be reversed by targeting OxPhos (Figure 7).

## Figures and Tables

**Figure 1 cells-09-02013-f001:**
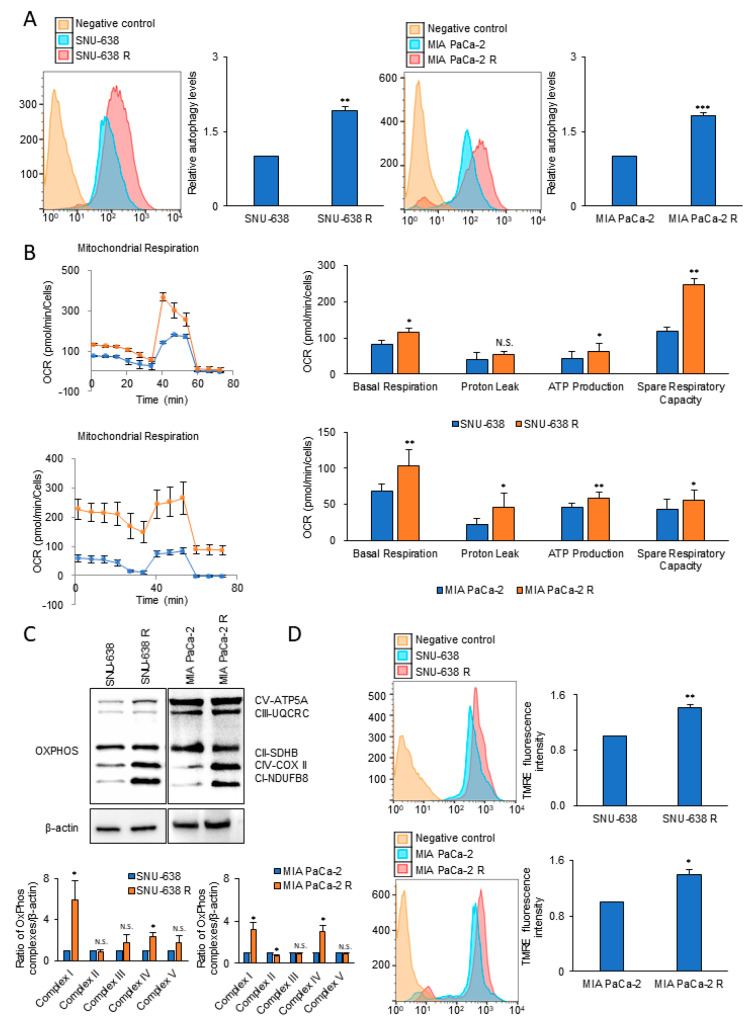
The levels of autophagy and OCR in irinotecan-resistant (R) cancer cells are increased compared with those in the wild-type control cells. (**A**) Autophagy levels were analyzed using Cyto-ID autophagy detection dye in irinotecan-resistant cancer cell lines compared with the wild-type counterpart (*n* = 3). (**B**) OCR and respiration parameters were measured by XFe96 extracellular flux analysis. OCR and ATP production were compared between irinotecan-resistant cancer cell lines and the wild-type counterparts (*n* = 3). (**C**) Levels of mitochondrial OxPhos complexes were analyzed by immunoblotting of wild-type and irinotecan-resistant lines of SNU-638 and MIA PaCa-2. (**D**) The mitochondrial membrane potential was analyzed by staining with TMRE in SNU-638, MIA PaCa-2, and their irinotecan-resistant lines (*n* = 3). Error bars represent the mean + s.d. *, *p* < 0.05; **, *p* < 0.01; ***, *p* < 0.001. n.s., no significant difference. *P* values were analyzed by unpaired two-tailed Student’s *t* test.

**Figure 2 cells-09-02013-f002:**
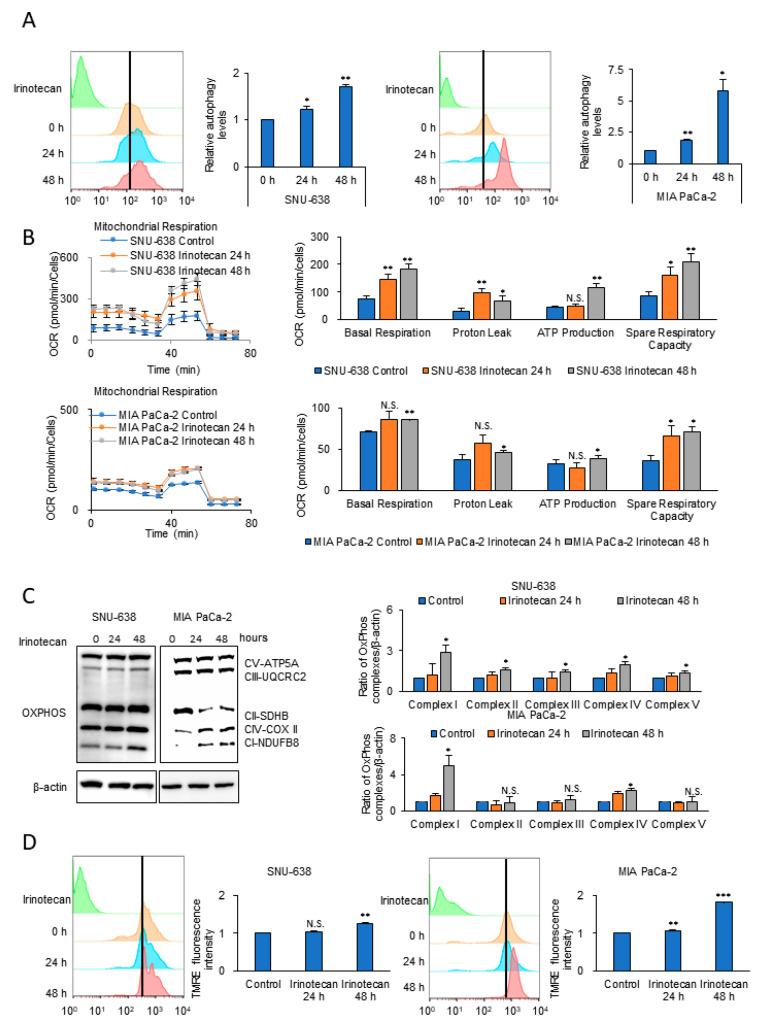
Anticancer drug treatment induces autophagy and OCR. (**A**) Autophagy levels were analyzed using Cyto-ID autophagy detection dye in SNU-638 and MIA PaCa-2 cells after irinotecan treatment for 24 and 48 h (*n* = 3). (**B**) OCRs and respiration parameters were measured by XFe96 extracellular flux analysis in SNU-638 and MIA PaCa-2 after irinotecan treatment for 24 and 48 h (*n* = 4). (**C**) Increased protein levels of OxPhos complexes were detected by immunoblotting after transient treatment of cancer cells with irinotecan for 24 and 48 h. The bands of the OxPhos components were quantified in relation to β-actin using ImageJ (*n* = 3). (**D**) The mitochondrial membrane potential in surviving SNU-638 and MIA PaCa-2 cells was analyzed by TMRE staining (*n* = 3). Error bars represent the mean + s.d. *, *p* < 0.05; **, *p* < 0.01; ***, *p* < 0.001. *P* values were analyzed by unpaired two-tailed Student’s *t* test.

**Figure 3 cells-09-02013-f003:**
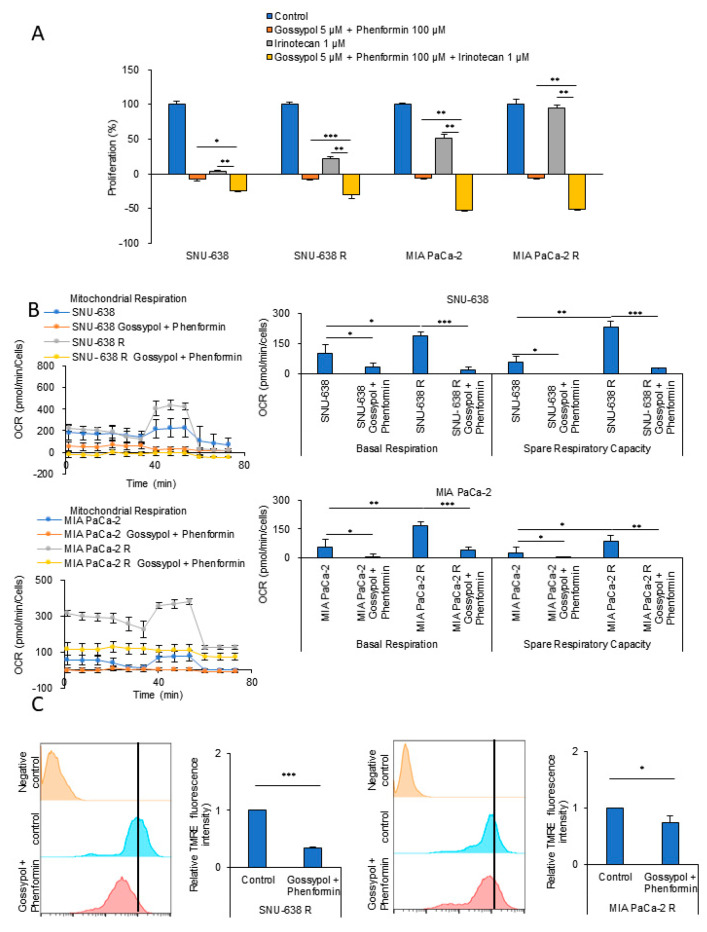
OxPhos inhibition reverses anticancer drug resistance. (**A**) Cell proliferation was assessed using the SRB assay. Wild-type (SNU-638, MIA PaCa-2) and drug-resistant cancer cells (SNU-638 R, MIA PaCa-2 R) were treated with the OxPhos inhibitors phenformin (100 μM) and gossypol (5 μM) for 48 h (*n* = 3). (**B**) The OCR was analyzed in wild-type and drug-resistant cancer cells after treatment with the OxPhos inhibitors phenformin (100 μM) and gossypol (5 μM) for 24 h (SNU-638; *n* = 3, MIA PaCa-2; *n* = 3). (**C**) The mitochondrial membrane potential was measured by TMRE staining in SNU-638 R and MIA PaCa-2 R cells before and after treatment with phenformin (100 µM) and gossypol (5 µM) for 24 h (*n* = 3). Error bars represent the mean + s.d. *, *p* < 0.05; **, *p* < 0.01; ***, *p* < 0.001. *P* values were analyzed by unpaired two-tailed Student’s t test.

**Figure 4 cells-09-02013-f004:**
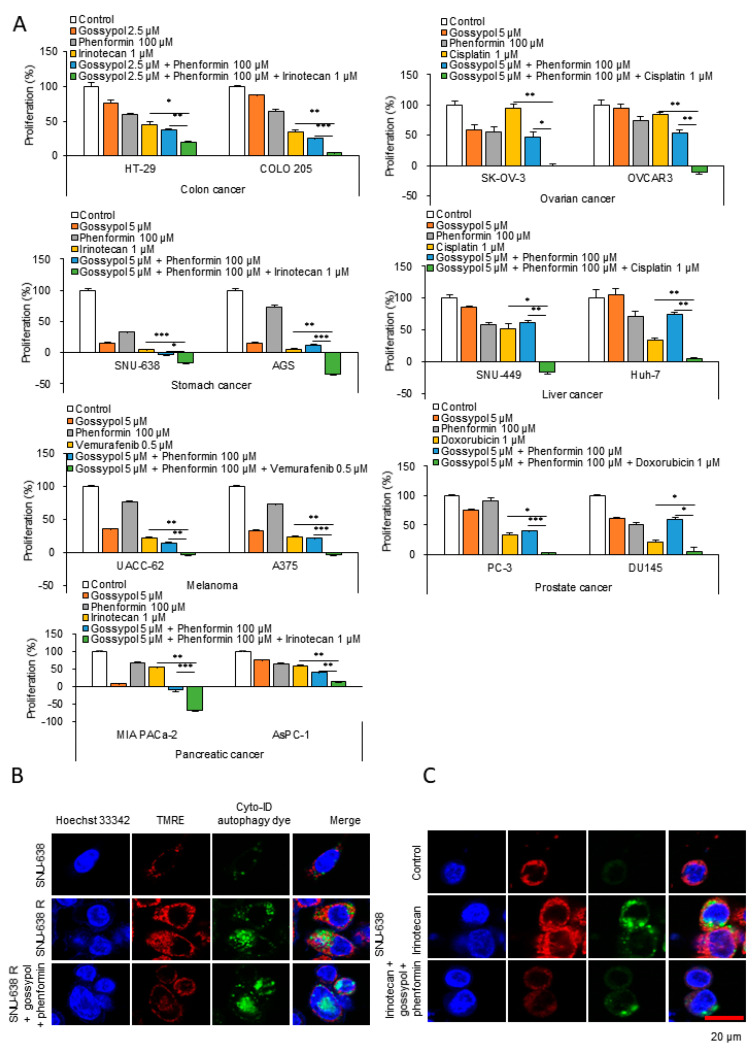
Anticancer drug treatment in combination with OxPhos inhibitors showed a synergistic effect in various cancer cell lines. (**A**) To test whether anticancer drugs with OxPhos inhibitors phenformin and gossypol worked synergistically, cell proliferation was measured using the SRB assay (*n* = 3). Colon, stomach, and pancreatic cancer cells were treated with 1 µM irinotecan for 48 h. Ovary and liver cancer cells were treated with 1 µM cisplatin for 48 h. Melanoma cells were treated with 0.5 mM vemurafenib for 48 h. Prostate cancer cells were treated with 1 µM doxorubicin for 48 h. For the synergy test, cells were treated with OxPhos inhibitors phenformin (100 µM) and gossypol (5 µM) for 48 h with or without anticancer drugs. (**B**) To compare levels of autophagy and OxPhos activity between wild-type and drug-resistant cancer cells, autophagy and the mitochondrial membrane potential were imaged using Cyto-ID autophagy detection dye and TMRE, respectively, in SNU638 and SNU-638R cells (*n* = 3). To test whether OxPhos inhibitors reduced the mitochondrial membrane potential without affecting autophagy, autophagy and the mitochondrial membrane potential were imaged in SNU-638R cells after phenformin and gossypol treatment for 6 h (*n* = 3). (**C**) To test whether autophagy and OxPhos were increased by anticancer drug treatment, autophagy and mitochondrial membrane potential were imaged in wild-type SNU-638 cells after treatment with 0.5 μM irinotecan for 48 h. To test whether OxPhos inhibitors reduced mitochondrial membrane potential without affecting autophagy, cells were treated with OxPhos inhibitors for 6 h. The levels of autophagy and mitochondrial membrane potential were determined by Cyto-ID and TMRE staining, respectively (*n* = 3). (**D**) To test whether irinotecan treatment induced mTORC1 activation leading to autophagy, phosphorylated forms of mTOR, p70S6K, 4E-BP1, and eIF4B were analyzed by immunoblotting after irinotecan treatment for 48 h in SNU-638 (*n* = 3). To test whether OxPhos inhibition blocks activation of the mTORC1 pathway, cells were treated with phenformin (100 µM) and gossypol (5 µM) for 6 h. Error bars represent the mean + s.d. *, *p* < 0.05; **, *p* < 0.01; ***, *p* < 0.001.

**Figure 5 cells-09-02013-f005:**
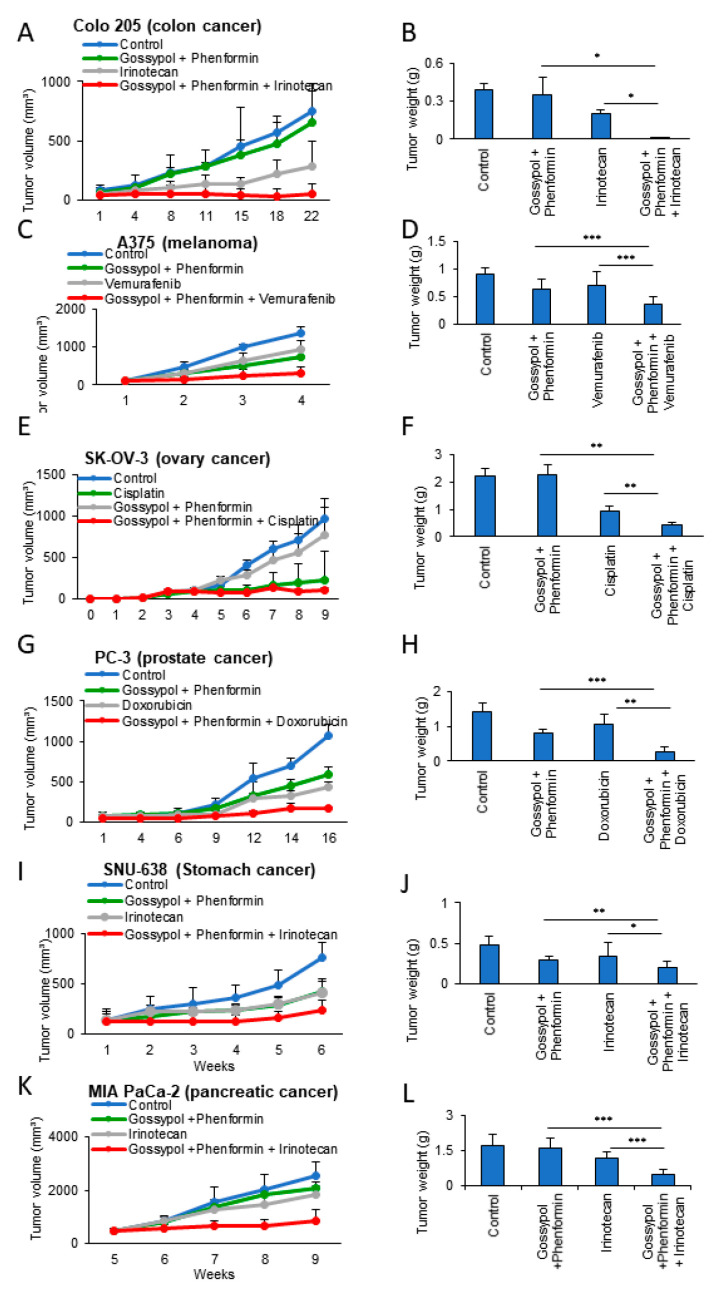
Synergistic inhibition of tumor xenografts by anticancer drugs with OxPhos inhibition. (**A**,**C**,**E**,**G**,**I**,**K**) Tumor volume measured using calipers. (**B**,**D**,**F**,**H**,**J**,**L**) Final weight of the subcutaneous tumors. (A, B) Colo 205 xenograft model with or without treatments (each group *n* = 3). (**C**,**D**) A375 xenograft model with or without treatments (*n* = 7). (**E**,**F**) SK-OV-3 xenograft model with or without treatments (*n* = 4). (**G**,**H**) PC-3 xenograft model with or without treatments (*n* = 4). (**I**,**J**) SNU-638 xenograft model with or without treatments (*n* = 5). (**K**,**L**) MIA PaCa-2 xenograft model with or without treatments (*n* = 5). Error bars represent the mean + s.d. *, *p* < 0.05; **, *p* < 0.01; ***, *p* < 0.001.

**Figure 6 cells-09-02013-f006:**
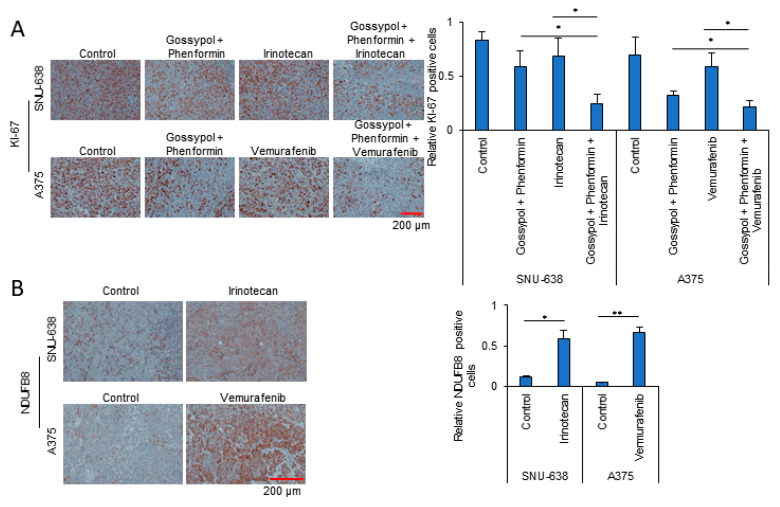
Synergistic anticancer effect of irinotecan treatment with OxPhos inhibitors. (**A**) Representative images showing immunohistochemical staining with anti-Ki-67 as a measure of tumor proliferative activity in the SNU-638 and A375 xenograft models used in Figure 5. (**B**) Representative images showing immunohistochemical staining with anti-NDUFB8 (mitochondrial complex I) as a measure of OxPhos activity in tumors from SNU-638 and A375 xenograft control and anticancer drug treatment groups. *P* values were analyzed by unpaired two-tailed Student’s t test. *, *p* < 0.05; **, *p* < 0.01.

**Figure 7 cells-09-02013-f007:**
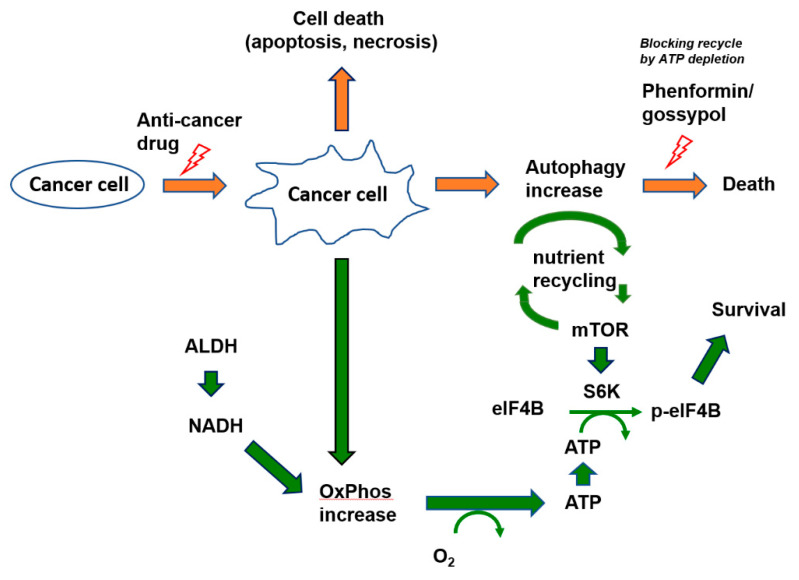
Schematic diagram of targeting cancer resistance by ATP depletion.

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
