# Peer review of "Targeting Oxidative Phosphorylation Reverses Drug Resistance in Cancer Cells by Blocking Autophagy Recycling"

_cells, 2020, doi:10.3390/cells9092013_

Round 1

Reviewer 1 Report

Authors have established the relationship between oxidative phosphorylation and autophagy in drug-resistant cancer cells. With results, it is amply clear that OxPhos in cancer cells, is regulated by ALDH, enhances the autophagy, which fuels the cancer cells. Thus, inhibition of the above process would become an attractive strategy in cancer.

The work is very interesting. I recommend the manuscript for publication. However, minor corrections that I am proposing would increase the quality of the manuscript.

Comments

Line 60: autophagy = autophagy enhancement.

Line 126: Expand 'ECL'

Line 127: Version of the Software should be provided.

Line 142: What is the logic behind the normalization of the OCR with the result of SRB assay; explain briefly.

Section 2.6 should be divided clearly into two procedures; flow cytometry and microscopy.

Line 174: 'before tumor induction' is redundant.

Line 195: Delete the 'comma' at the end.

Please provide the established reference for the resistant cancer cells generation.

Line 217: Short form would be okay here; 'OCR'.

Line 216-234: Be sure that the discussion is about establishing resistance. It means after several cycles of anti-cancer drug treatment, both autophagy and OxPhos have increased. Not with 24-48 h treatment; as irinotecan is an anti-cancer drug, which induces loss of mitochondrial membrane potential. 

Figure 1 legend: Please include the name of the statistical test performed, although it is available in procedures.

Figure 2: Which type of statistical tests have performed for the comparison of "control, irinotecan 24 h, and irinotecan 48 h.

Figure 3 looks blurred; please improve the resolution.

Figure 4 should be enlarged, or resolution should be increased for better visibility.

Figure 6 excel graphs are bit blurred; please increase the resolution.

Author Response

Reviewer 1

Line 60: autophagy = autophagy enhancement.

Thank you. It is fixed to autophagy enhancement.

Line 126: Expand 'ECL'

Thank you. It is fixed to enhanced chemiluminescent (ECL).

Line 127: Version of the Software should be provided.

Thank you. We added a version of the software: (64-bit Java 1.8.0_112)

Line 142: What is the logic behind the normalization of the OCR with the result of SRB assay; explain briefly.

Thank you. We explained that

“Finally, the OCR data were normalized to account for the differences in the initial cell numbers for each well by using the SRB assay.”

Section 2.6 should be divided clearly into two procedures; flow cytometry and microscopy.

Thank you. We divided experimental methods into 2 paragraphs.

Line 174: 'before tumor induction' is redundant.

Thank you. We deleted 'before tumor induction'

Line 195: Delete the 'comma' at the end.

Thank you. It is fixed.

Please provide the established reference for the resistant cancer cells generation.

- Thank you. We added a reference.

Reference: doi: 10.1371/journal.pone.0054193 (Generation and Characterisation of Cisplatin-Resistant Non-Small Cell Lung Cancer Cell Lines Displaying a Stem-Like Signature).

Line 217: Short form would be okay here; 'OCR'.

Thank you. It is fixed to OCR.

Line 216-234: Be sure that the discussion is about establishing resistance. It means after several cycles of anti-cancer drug treatment, both autophagy and OxPhos have increased. Not with 24-48 h treatment; as irinotecan is an anti-cancer drug, which induces loss of mitochondrial membrane potential. 

Thank you. It is a good comment. We changed “drug-resistant” to “drug-treated” at 234 line.

Figure 1 legend: Please include the name of the statistical test performed, although it is available in procedures.

Figure 2: Which type of statistical tests have performed for the comparison of "control, irinotecan 24 h, and irinotecan 48 h.

Thank you. We added the name of the statistical test.

Figure 3 looks blurred; please improve the resolution.

Thank you. We improved the resolution.

Figure 4 should be enlarged, or resolution should be increased for better visibility.

Thank you. We enlarged the figure.

Figure 6 excel graphs are bit blurred; please increase the resolution.

Thank you. We improved the resolution.

Reviewer 2 Report

The manuscript “Targeting Oxidative Phoshorylation Reverses Drug Resistance in Cancer Cells by Blocking Autophagy Recycling” is dealing with really up-to-date topic of drug resistance and its mechanisms.

The manuscript is written and designed well. However, the manuscript has a few shortcomings. In the Introduction, the facts are lacking the citations (lines 57-59), please, add at least two of them. In the Materials and Methods, part 2.3. Western Blot, lines 130-134, there are in many places brackets immediately after the words without space, f.e. mTOR(Ser2448) instead of mTOR (Ser2448). The manuscript contains few grammatical and stylistic errors.

This study is interesting and has an impact on the scientific community. After minor revisions, I would highly recommend this manuscript to be accepted to the Cells.

Author Response

Reviewer 2

The manuscript is written and designed well. However, the manuscript has a few shortcomings. In the Introduction, the facts are lacking the citations (lines 57-59), please, add at least two of them.

- Thank you. We added two references.

Reference: doi: 10.1158/1078-0432.CCR-09-1070 (Resistance to Targeted Therapies: Refining Anticancer Therapy in the Era of Molecular Oncology)

doi: 10.1016/j.molonc.2014.05.004 (Drug resistance to targeted therapies: Déjà vu all over again)

In the Materials and Methods, part 2.3. Western Blot, lines 130-134, there are in many places brackets immediately after the words without space, f.e. mTOR(Ser2448) instead of mTOR (Ser2448). The manuscript contains few grammatical and stylistic errors.

- Thank you. We have fixed them all.